# DNA methylation signatures in peripheral blood strongly predict all-cause mortality

Yan Zhang[1], Rory Wilson[2,3], Jonathan Heiss[1], Lutz P. Breitling[1], Kai-Uwe Saum[1], Ben Schöttker[1,4],
Bernd Holleczek[5], Melanie Waldenberger[2,3], Annette Peters[2,3] & Hermann Brenner[1,6,7]

DNA methylation (DNAm) has been revealed to play a role in various diseases. Here we performed epigenome-wide screening and validation to identify mortality-related DNAm signatures in a general population-based cohort with up to 14 years follow-up. In the discovery panel in a case-cohort approach, 11,063 CpGs reach genome-wide significance (FDR < 0.05). 58 CpGs, mapping to 38 well-known disease-related genes and 14 intergenic regions, are confirmed in a validation panel. A mortality risk score based on ten selected CpGs exhibits strong association with all-cause mortality, showing hazard ratios (95% CI) of 2.16 (1.10–4.24), 3.42 (1.81–6.46) and 7.36 (3.69–14.68), respectively, for participants with scores of 1, 2–5 and 5+ compared with a score of 0. These associations are confirmed in an independent cohort and are independent from the 'epigenetic clock'. In conclusion, DNAm of multiple disease-related genes are strongly linked to mortality outcomes. The DNAm-based risk score might be informative for risk assessment and stratification.

[1] Division of Clinical Epidemiology and Aging Research, German Research Cancer Research Center (DKFZ), Im Neuenheimer Feld 280, D-69120 Heidelberg, Germany. [2] Research Unit of Molecular Epidemiology, Helmholtz Zentrum München, German Center for Environmental Health, D-85764 Neuherberg, Germany. [3] Institute of Epidemiology II, Helmholtz Zentrum München, German Research Center for Environmental Health, D-85764 Neuherberg, Germany. [4] Network Ageing Research, University of Heidelberg, Bergheimer Strasse 20, D-69115 Heidelberg, Germany. [5] Saarland Cancer Registry, Präsident Baltz Strasse 5, D-66119 Saarbrücken, Germany. [6] Division of Preventive Oncology, German Cancer Research Center (DKFZ) and National Center for Tumor Diseases (NCT), Im Neuenheimer Feld 460, D-69120 Heidelberg, Germany. [7] German Cancer Consortium (DKTK), German Cancer Research Center (DKFZ), Im Neuenheimer Feld 280, D-69120 Heidelberg, Germany. Correspondence and requests for materials should be addressed to Y.Z. (email: y.zhang@dkfz.de).

DNA methylation (DNAm), as the most widely studied form of epigenetic programming, has been revealed to be modulated by lifestyle and environmental factors[1,2] and to be involved in onset and progression of complex diseases, including various forms of malignant diseases, cardiovascular diseases (CVDs), metabolic diseases (for example, diabetes), neuropsychiatric disorders and autoimmune disorders[3–7]. Therefore, DNAm could plausibly be associated with the excess mortality from specific diseases and consequently with all-cause mortality. This was exemplified by the previous investigations on smoking-associated DNAm changes and their relationship with lung cancer incidence/mortality and mortality from any cause, cancer and CVD[8–10].

In addition, evidence has accumulated that the recently established 'epigenetic clock' (also known as DNAm age) based on age-associated DNAm changes, which presumably reflects individuals' biological age, is indicative for ageing-related outcomes and longevity[11–14]. Following the first study reporting an association of DNAm age with all-cause mortality by Marioni et al.[13], the association was consistently demonstrated in various longitudinal studies[15,16], for individual age-associated CpGs[17] and also for newly identified age-associated CpGs[18]. On the other hand, several epigenome-wide association studies (EWASs) have pointed out that DNAm involved in ageing-related phenotypes are largely distinct from the established age-associated DNAm[19–21].

To unravel the determinants of survival in the DNAm landscape, we performed an epigenome-wide screening and replication for mortality-related DNAm signatures in a general population-based cohort of older adults. Here we show that DNAm of 58 CpGs in baseline blood samples are associated with mortality from any causes during 14 years of follow-up. A mortality risk score based on ten selected CpGs strongly predicts all-cause, CVD and cancer mortality, also in an independent population-based cohort. The identified DNAm markers may thus bear implications in risk assessment and stratification in clinical practice.

## Results

**Study population**. Table 1 presents the baseline characteristics of the ESTHER (Epidemiologische Studie zu Chancen der Verhütung, Früherkennung und optimierten Therapie chronischer Erkrankungen in der älteren Bevölkerung) study population. Of the 406 deaths in the case–cohort sample of the discovery panel, 90 were also included in the subcohort owing to random selection of subcohort at baseline. The time between blood sample collection and death ranged from 0.2 to 12.3 years (median (interquartile range (IQR), 7.4 (4.5–9.6) years) for these 406 participants. The corresponding figures for the 231 deaths in the validation panel were 0.2–13.8 years (range) and 8.6 (5.6–11.6) years (median (IQR)). The characteristics of the participants in the subcohort of the discovery panel are similar as those of the participants in the validation panel, except that the proportion of women was larger in the subcohort than in the validation cohort. In comparison with those two subgroups, the group of deceased participants in the discovery panel featured higher proportions of men, smokers, old (>70 years) and inactive participants, and participants with prevalence of hypertension, diabetes, CVD and cancer at baseline. The characteristics of the KORA (Kooperative Gesundheitsforschung in der Region Augsburg) study population are presented in Supplementary Table 1. The average age was similar in KORA and ESTHER participants (61 versus 62 years), but KORA participants had a much broader age range (31–82 years) than ESTHER participants (50–75 years).

**Discovery and validation of mortality-related CpGs**. In the discovery phase, a total of 11,063 CpGs passed the genome-wide significance threshold (false discovery rate (FDR) <0.05) (Supplementary Fig. 1). Associations with all-cause mortality were successfully replicated for 58 CpGs even after comprehensive confounder adjustment in the validation phase. Manhattan plots for the discovery and validation analyses are presented in Supplementary Fig. 2. Table 2 shows the results for the 58 CpGs. Methylation at the vast majority (49 of 58 CpGs) was inversely associated with mortality, with hazard ratios (HRs) and 95% confidence intervals (95% CIs) for a decrease in methylation by 1 s.d. ranging from 1.16 (1.04–1.28) to 1.95 (1.29–2.94). HRs (95% CI) for the other 9 CpGs showing positive associations with mortality ranged from 0.60 (0.47–0.77) to 0.83 (0.71–0.97) per s.d. decrease in methylation. The 58 loci are located at 38 genes and 14 intergenic regions across 19 chromosomes. In addition to three CpGs within AHRR, ten clusters within the identified sites were observed (Table 2), that is, 1p21.2 (2 CpGs), 2q37 (2 CpGs), 3q11/12 (2 CpGs), 6p21 (4 CpGs), 11p15 (3 CpGs), 11q13 (3 CpGs), 17q21 (2 CpGs), 17q25 (2 CpGs), 19p13 (3 CpGs) and 19q13 (7 CpGs). A literature search in PubMed for genes containing the identified CpGs found evidence that these genes or their methylation are involved in a variety of major diseases, including diabetes (for example, SARS, SQLE, NFE2L3, KCNQ1OT1 and SOCS3), CVD (for example, SARS, VCAM1, PLCL2, UTS2D, AHRR, 6p21.33, SQLE, KCNQ1OT1, SEMA7A, F2RL3, BCL3, PPP1R15A, PDE9A and MIR19A), various forms of cancers (for example, SOCS3, SLC1A5, MIR19A, MIR10A, CALR, ERCC1, BCL3, SQLE, RARA, LAPTM5, INPP5A, CSGALNACT1, KCNQ1OT1, CDC42BPB, PDE9A and MKL1), neuropsychiatric disorders (FOSL2, ATL3, SHANK2 and PPP1R15A) and HIV infection (for example, GPR15 and MIR10A) (Table 2 and Supplementary Table 2). Several of those genes, such as SQLE, KCNQ1OT1 and SOCS3, have been suggested to play roles in multiple types of diseases. Means and s.d. of the 58 CpGs at baseline among deceased participants and survivors are illustrated in Fig. 1.

**Associations of risk factors with mortality-related CpGs**. In the analyses of associations between the 58 CpGs and the covariates, differences in methylation levels with respect to age and sex were observed for 23 and 25 CpGs, respectively (Supplementary Table 3). However, none of the 58 CpGs overlapped with previously identified ageing-related sites[11,12,18,22,23]. Forty-eight of the 58 CpGs were differentially methylated according to smoking exposure and 22 of the 48 CpGs had also been found to be associated with smoking by previous EWASs[2,24] (CpGs displayed in bold in Table 2). Five of the 48 smoking-associated CpGs and cg24397007 in FOSL2 were also associated with alcohol consumption (Supplementary Table 3). Four of the 48 smoking-associated CpGs and cg08362785 in MKL1 were also associated with prevalent diabetes; of these 5 sites, cg18181703 in SOCS3 was also recently identified to be associated with type 2 diabetes (T2D)[5,25] and cg23190089 is located at SLC22A18AS, a locus near to known methylation-regulated genes implicated in T2D[26,27]. In addition, 4 of the 48 smoking-associated CpGs, including 2 diabetes-associated sites (cg18181703 in SOCS3 and cg26470501 in BLC3), were also associated with prevalent cancer. An illustration of the 48 CpGs is presented in Supplementary Fig. 3.

**Mortality risk score and validation**. Ten CpGs (cg01612140, cg05575921, cg06126421, cg08362785, cg10321156, cg14975410, cg19572487, cg23665802, cg24704287 and cg25983901) were selected by least absolute shrinkage and selection operator

**Table 1 | Characteristics of study population at baseline.**

| Characteristics | Discovery panel N (%) | | Validation panel N (%) |
| --- | --- | --- | --- |
| | All deaths (n = 406) | Subcohort (n = 548)* | Cohort (n = 1,000) |
| *Sex* | | | |
| Male | 224 (55.2) | 212 (38.7) | 500 (50.0) |
| Female | 182 (44.8) | 336 (61.3) | 500 (50.0) |
| *Age (years)* | | | |
| 50–60 | 84 (20.7) | 179 (32.7) | 339 (33.9) |
| 60–64 | 97 (23.9) | 159 (29.0) | 289 (28.9) |
| 65–69 | 113 (27.8) | 127 (23.2) | 226 (22.6) |
| 70–75 | 112 (27.6) | 83 (15.1) | 146 (14.6) |
| *Smoking status[†]* | | | |
| Never smoker | 155 (39.6) | 251 (47.3) | 469 (48.0) |
| Former smoker | 136 (34.8) | 180 (33.9) | 323 (33.0) |
| Current smoker | 100 (25.6) | 100 (18.8) | 186 (19.0) |
| *Body mass index (kg m$^{-2}$)[‡]* | | | |
| Underweight (<18.5) | 5 (1.2) | 1 (0.2) | 8 (0.8) |
| Normal weight (18.5 to <25.0) | 117 (28.9) | 166 (30.3) | 243 (24.4) |
| Overweight (25.0 to <30.0) | 173 (42.7) | 235 (42.9) | 483 (48.5) |
| Obesity (≥30.0) | 110 (27.2) | 146 (26.6) | 263 (26.4) |
| *Physical activity[§]* | | | |
| Inactive | 108 (26.7) | 114 (20.9) | 203 (20.3) |
| Low | 205 (50.6) | 268 (49.1) | 438 (43.8) |
| Medium/high | 92 (22.7) | 164 (30.0) | 358 (35.8) |
| *Prevalence of major diseases* | | | |
| Hypertension | 278 (68.5) | 308 (56.2) | 589 (58.9) |
| Diabetes[‖] | 108 (26.6) | 95 (17.4) | 162 (16.2) |
| CVD[‖] | 120 (29.6) | 97 (17.7) | 182 (18.2) |
| Cancer | 57 (14.0) | 27 (4.9) | 66 (6.6) |

CVD, cardiovascular disease.
*The subcohort included 90 deaths due to random selection at baseline irrespective of death status during follow-up.
†Data missing for 27 and 22 subjects, respectively, in discovery and validation panels.
‡Data missing for one and three subjects, respectively, in discovery and validation panels.
§Data missing for three and one subjects, respectively, in discovery and validation panels.
‖Data missing for one subject in both discovery and validation panels.

(LASSO) regression. Preliminary analyses in ESTHER samples showed that ≥40% deaths occurred among participants with methylation levels in the highest quartile of cg08362785 (hyper-methylated among deaths) or in the first quartile of the other 9 CpGs (demethylated among deaths) (Supplementary Fig. 4a). We therefore used the fourth quartile value of cg08362785 and first quartile values of other nine CpGs as the cutoff points, to define aberrant methylation for each CpG (the exact cutoff points are listed in Supplementary Table 4). Participants with aberrant methylation at 1–10 CpGs had a mortality score of 1–10, respectively, and participants without aberrant methylation at any of the 10 CpGs had score of 0. Table 3 shows the associations of score with all-cause mortality. Compared with participants with a score of 0, those who had a score of 1, 2–5 and 5+ had 2-, 3- and 7-fold risk of dying, controlling for all the potential confounding factors. Analyses restricted to only older participants (≥60 years) yielded essentially the same risk estimates, for example, HRs (95% CI) were 2.14 (1.02–4.47), 3.38 (1.68–6.80) and 7.44 (3.50–15.84), respectively, for a score of 1, 2–5 and 5+ versus score = 0. Similar patterns of distribution of deceased were also observed in KORA participants (Supplementary Fig. 4b). Using the cutoff points from the ESTHER cohort defining aberrant methylation of ten CpGs (Supplementary Table 4), replicated analyses in the KORA cohort showed consistent patterns and similar risk estimates (Table 3). Crude HRs (95% CI) for participants with score of 1, 2–5 and 5+ were 1.21 (0.37–3.97), 6.42 (2.55–16.18) and

19.29 (5.58–66.63), respectively, compared with score = 0. In the fully adjusted model, three- and six-fold increases in mortality persisted for score levels of 2–5 and 5+, respectively. Using cutoff points (quartiles) of KORA itself defining aberrant methylation of the ten CpGs to build the mortality score, risk estimates were larger than those derived from using ESTHER's cutoff points. For example, the HR (95% CI) in the fully adjusted model was 7.41 (1.61–34.07) for participants with score of 5+. In addition, a continuous risk score was computed through linear combination of LASSO regression coefficient weighted methylation values of the ten CpGs (the combination formula is presented in Supplementary Fig. 1). A similar trend that mortality monotonously increased with increasing continuous risk score was observed in both the ESTHER (risk score ranged from −3.92 to −0.72; median (IQR), −2.70 (−2.98 to −2.35)) and the KORA cohorts (risk score ranged from −4.40 to −1.51; median (IQR), −3.15 (−3.41 to −2.86)). Figure 2 shows the corresponding dose–response relationships derived from restricted cubic spline regression with adjustment for all the covariates again[28].

Sex-specific analyses indicated the associations with all-cause mortality to be stronger among women than among men in both cohorts (Supplementary Table 5). Table 4 shows that the associations of score with CVD mortality were stronger than with cancer mortality in both cohorts. The corresponding survival curves in the ESTHER cohort are presented in Fig. 3. Similar

**Table 2 | Association of 58 CpGs with all-cause mortality in the validation panel.**

| CpG site | HR (95% CI)* | FDR | Chr position (GRCh37/hg19) | Gene name | Gene-related major diseases[†] |
|---|---|---|---|---|---|
| **cg03725309** | 1.34 (1.10–1.62) | 0.0450 | 1p13.3 (chr1:109757585) | SARS | T2D (M); coronary artery disease |
| cg25763716 | 1.29 (1.02–1.63) | 0.0486 | **1p21.2 (chr1:101184304)** | VCAM1 | atherosclerosis; MI; tumour invasion |
| cg13854219 | 1.51 (1.05–2.17) | 0.0399 | **1p21.2 (chr1:101757037)** | | |
| **cg25189904** | 1.18 (1.02–1.38) | 0.0450 | 1p31.3 (chr1:68299493) | GNG12 | Endometrial cancer |
| cg15459165 | 0.60 (0.47–0.77) | 0.0035 | 1p35.2 (chr1:31223850) | LAPTM5 | Lung cancer (M); NB (M); multiple myeloma (M) |
| cg19266329 | 1.33 (1.14–1.55) | 0.0179 | 1q21.1 (chr1:145456128) | | |
| cg24397007 | 1.28 (1.08–1.53) | 0.0483 | 2p23.2 (chr2:28619095) | FOSL2 | Parkinson's disease (M); breast cancer |
| **cg23079012** | 1.16 (1.04–1.28) | 0.0008 | 2p25.1 (chr2:8343711) | | |
| **cg27241845** | 1.23 (1.06–1.44) | 0.0222 | **2q37.1 (chr2:233250371)** | | |
| cg06905155 | 1.20 (1.05–1.36) | 0.0450 | **2q37.3 (chr2:240723946)** | | |
| cg16503724 | 0.77 (0.64–0.94) | 0.0484 | 3p24.3 (chr3:17130667) | PLCL2 | Renal cell carcinoma (M); MI; systemic sclerosis |
| **cg19859270** | 1.32 (1.13–1.54) | 0.0001 | **3q11.2 (chr3:98251295)** | GPR15 | HIV |
| **cg02657160** | 1.22 (1.07–1.38) | 0.0084 | **3q12.1 (chr3:98311063)** | CPOX | |
| cg14975410 | 1.20 (1.04–1.38) | 0.0372 | 3q26.31 (chr3:171180070) | | |
| cg14855367 | 1.23 (1.08–1.40) | 0.0463 | 3q28 (chr3:191048309) | UTS2D | Coronary artery disease |
| **cg05575921** | 1.51 (1.25–1.84) | 4.25E−07 | **5p15.33 (chr5:373378)** | AHRR | Lung cancer (M); atherosclerosis (M) CVD/cancer death (M) |
| **cg14817490** | 1.19 (1.01–1.42) | 0.0260 | **5p15.33 (chr5:392920)** | AHRR | |
| **cg21161138** | 1.23 (1.05–1.44) | 3.07E−05 | **5p15.33 (chr5:399361)** | AHRR | |
| **cg12513616** | 1.17 (1.01–1.36) | 0.0280 | 5q35.3 (chr5:177370977) | | |
| cg20732076 | 1.25 (1.05–1.50) | 0.0217 | **6p21.1 (chr6:42335232)** | TRERF1 | Breast cancer |
| cg25285720 | 1.25 (1.06–1.46) | 0.0488 | **6p21.32 (chr6:32919434)** | HLA-DMA | Ovarian cancer (M) |
| **cg06126421** | 1.33 (1.10–1.60) | 0.0008 | **6p21.33 (chr6:30720081)** | | Lung cancer (M); CVD/cancer death (M) |
| **cg15342087** | 1.17 (1.01–1.36) | 0.0450 | **6p21.33 (chr6:30720210)** | | |
| cg01612140 | 1.40 (1.14–1.72) | 0.0244 | 6q14.1 (chr6:78166437) | | |
| cg25983901 | 1.19 (1.02–1.40) | 0.0450 | 7p12.3 (chr7:46972700) | | |
| cg12510708 | 1.33 (1.06–1.67) | 0.0241 | 7p15.2 (chr7:26193806) | NFE2L3 | T2D (M); breast cancer (M) |
| cg26286961 | 1.27 (1.10–1.47) | 0.0260 | 8p21.3 (chr8:19460209) | CSGALNACT1 | FV-PTC; multiple myeloma |
| cg00285394 | 1.20 (1.05–1.36) | 0.0217 | 8q24.13 (chr8:126011954) | SQLE | T2D/CVD (M); breast cancer (M); lung/prostate cancer |
| cg01140244 | 0.69 (0.54–0.89) | 0.0450 | 10q26.3 (chr10:134498960) | INPP5A | Brain tumour; cutaneous squamous cell carcinoma |
| cg23190089 | 1.40 (1.08–1.82) | 0.0450 | **11p15.4 (chr11:2920209)** | SLC22A18AS | Breast cancer(M) |
| **cg07123182** | 1.26 (1.11–1.44) | 0.0003 | **11p15.5 (chr11:2722391)** | KCNQ1OT1 | T2D (M); CRC (M); MI; breast cancer |
| **cg26963277** | 1.31 (1.14–1.49) | 3.07E−05 | **11p15.5 (chr11:2722408)** | KCNQ1OT1 | |
| cg18550212 | 1.57 (1.22–2.01) | 0.0217 | **11q13.1 (chr11:63435428)** | ATL3 | Neuropathy |
| cg10321156 | 1.20 (1.02–1.42) | 0.0450 | **11q13.1 (chr11:63687223)** | | |
| cg25193885 | 0.78 (0.65–0.93) | 0.0100 | **11q13.3 (chr11:70328867)** | SHANK2 | Prostate cancer (M); neuropsychiatric disorders |
| **cg07986378** | 1.27 (1.03–1.57) | 0.0483 | 12p13.2 (chr12:11898285) | ETV6 | Haematopoiesis and malignant transformation |
| cg23665802 | 1.39 (1.13–1.71) | 0.0122 | 13q31.3 (chr13:92002338) | MIR19A | CVD; lung/gastric/breast/bladder/cervical cancer/CRC/HCC |
| cg04987734 | 0.81 (0.70–0.94) | 0.0266 | 14q32.32 (chr14:103415874) | CDC42BPB | Tumour cell invasion, for example, CRC (M); breast cancer |
| cg19459791 | 0.83 (0.71–0.97) | 0.0483 | 15q22.31 (chr15:65363023) | | |
| **cg00310412** | 1.26 (1.07–1.47) | 0.0241 | 15q24.1 (chr15:74724919) | SEMA7A | Multiple sclerosis; lung/liver fibrosis |
| cg26709988 | 1.95 (1.29–2.94) | 0.0092 | 16q24.1 (chr16:84860919) | CRISPLD2 | |
| cg23842572 | 0.75 (0.62–0.91) | 0.0194 | 17p11.2 (chr17:17030253) | MPRIP | Cancer cell invasion |
| **cg19572487** | 1.26 (1.07–1.49) | 0.0003 | **17q21.2 (chr17:38476025)** | RARA | Breast cancer (M); hepatocellular/thyroid carcinomas (M) |
| cg01572694 | 1.36 (1.12–1.67) | 0.0311 | **17q21.32 (chr17:46657555)** | MIR10A | Lung/gastric/breast/colon/pancreatic/brain cancer/HCC; HIV |
| cg08546016 | 1.39 (1.13–1.70) | 0.0372 | **17q25.1 (chr17:72776239)** | TMEM104 | |
| cg18181703 | 1.24 (1.07–1.44) | 0.0214 | **17q25.3 (chr17:76354622)** | SOCS3 | T2D (M); lung/pancreatic/cervical/endometrial/prostate cancer/HNSCC/HCC/CRC/melanoma/glioblastoma/leukaemia (M) |
| **cg03636183** | 1.27 (1.06–1.51) | 0.0003 | **19p13.11 (chr19:17000586)** | F2RL3 | Lung cancer (M); CVD/cancer death (M) |
| cg24704287 | 1.31 (1.06–1.61) | 0.0329 | **19p13.13 (chr19:13951482)** | | |
| cg11341610 | 1.29 (1.04–1.59) | 0.0421 | **19p13.2 (chr19:13050932)** | CALR | Lung/gastric/pancreatic/prostate/ovarian cancers/NB |
| cg14085840 | 1.45 (1.10–1.91) | 0.0486 | **19q13.2 (chr19:40939429)** | | |
| **cg26470501** | 1.20 (1.04–1.39) | 0.0351 | **19q13.32 (chr19:45252955)** | BCL3 | CVD; lung/breast/prostate cancer/CRC |
| **cg05492306** | 1.39 (1.07–1.80) | 0.0217 | **19q13.32 (chr19:45927594)** | ERCC1 | Lung/breast cancer(M); HNSCC/gastric cancer |
| cg25607249 | 1.41 (1.10–1.81) | 0.0345 | **19q13.32 (chr19:47288040)** | SLC1A5 | T2D (M); lung/pancreatic/breast/prostate cancer/CRC/NB/melanoma/renal cell carcinoma |
| cg01406381 | 1.52 (1.19–1.95) | 0.0054 | **19q13.32 (chr19:47288263)** | SLC1A5 | |
| cg07626482 | 1.19 (1.03–1.38) | 0.0463 | **19q13.32 (chr19:47289503)** | SLC1A5 | |
| **cg03707168** | 1.29 (1.02–1.63) | 0.0311 | **19q13.33 (chr19:49379127)** | PPP1R15A | Neurological diseases; myocardial ischaemia |
| cg25491402 | 0.65 (0.47–0.90) | 0.0496 | 21q22.3 (chr21:44101491) | PDE9A | Lung cancer (M); CVD; breast cancer |
| cg08362785 | 0.63 (0.51–0.78) | 0.0003 | 22q13.1 (chr22:40814879) | MKL1 | Lung/breast cancer(M); lung/liver fibrosis (M) |

CI, confidence interval; CVD, cardiovascular disease; FDR, false discovery rate; FV-PTC, follicular variant of papillary thyroid carcinoma; HCC, hepatocellular carcinoma; HIV, human immunodeficiency virus; HNSCC, head and neck squamous cell carcinoma; HR, hazard ratio; MI, myocardial infarction; NB, neuroblastoma; T2D, type 2 diabetes.
Bold printed CpGs ($n = 22$) are sites identified to be associated with smoking in both the current and previous epigenome-wide association studies. Underscored CpGs ($n = 10$) were selected to develop the mortality risk score. Bold printed 'Chr position' indicates clusters of identified CpGs.
*HRs for a decrease in methylation by 1 s.d.; model adjusted for age, sex, smoking status, body mass index, physical activity, systolic blood pressure, total cholesterol, hypertension and prevalent cardiovascular disease, diabetes and cancer at baseline.
†M refers to diseases, which have been reported to be related to methylation of the gene; detailed descriptions of gene function and relevant diseases are listed in Supplementary Table 1.

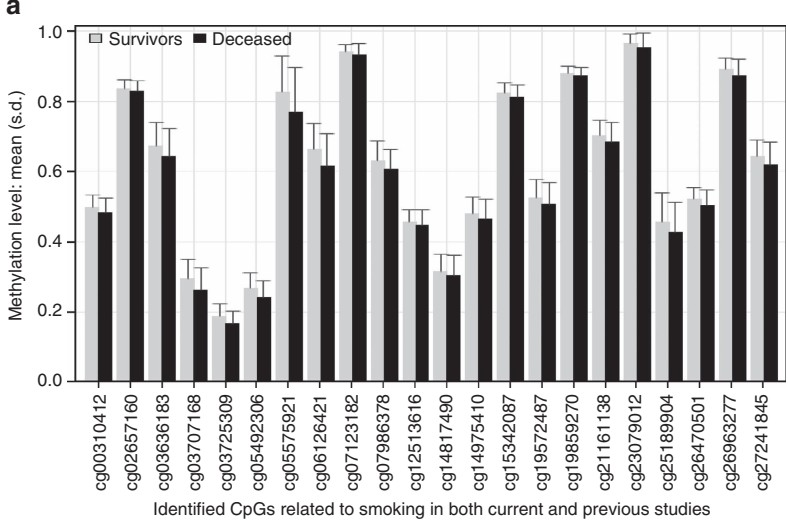

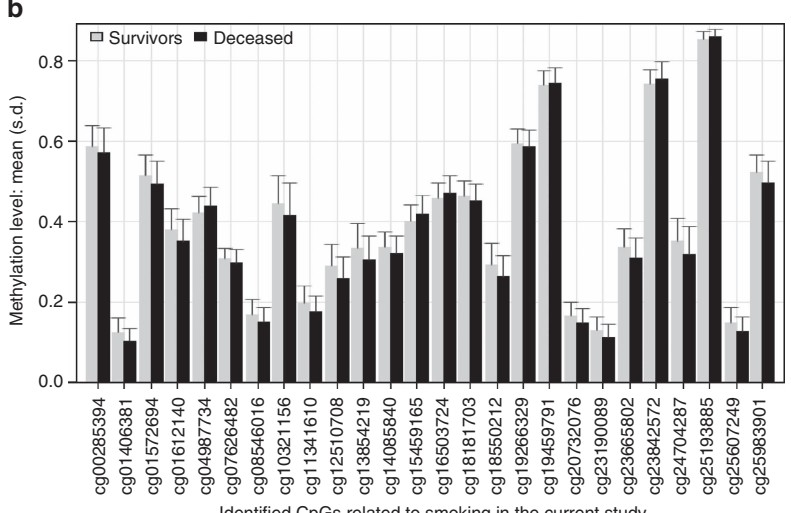

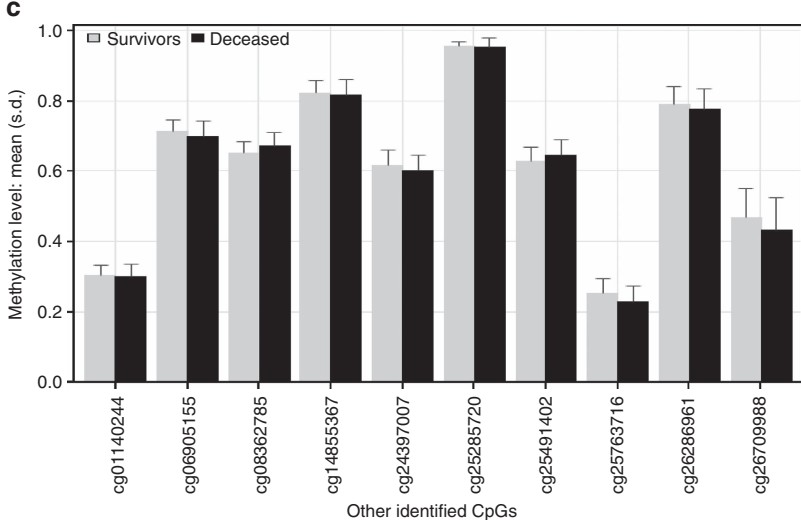

**Figure 1 | Methylation levels of 58 CpGs among deceased ($N = 231$) and survivors ($N = 769$) in the validation panel of the ESTHER cohort.** (**a**) Mean and s.d. (error bar) of 22 mortality-related CpGs (also discovered to be associated with smoking in both current and previous studies) by vital status; (**b**) mean and s.d. (error bar) of 26 mortality-related CpGs (also discovered to be associated with smoking in the current study) by vital status; (**c**) mean and s.d. (error bar) of other 10 mortality-related CpGs by vital status.

**Table 3 | Association of the risk score with all-cause mortality in the ESTHER and KORA study.**

| Study | Mortality score* | $N_{total}$ | Cases | PY | IR† | HR (95% CI) | | |
| | | | | | | Model 1‡ | Model 2§ | Model 3‖ |
|---|---|---|---|---|---|---|---|---|
| ESTHER study | 0 | 199 | 14 | 2690.69 | 0.52 | Ref. | Ref. | Ref. |
| | 1 | 242 | 41 | 3144.50 | 1.30 | 2.55 (1.39–4.68) | 2.04 (1.11–3.75) | 2.16 (1.10–4.24) |
| | 2–5 | 426 | 105 | 5300.86 | 1.98 | 3.93 (2.25–6.86) | 3.18 (1.81–5.59) | 3.42 (1.81–6.46) |
| | >5 | 131 | 70 | 1348.99 | 5.19 | 10.89 (6.13–19.35) | 7.64 (4.21–13.85) | 7.36 (3.69–14.68) |
| KORA study | 0 | 487 | 5 | 2163.01 | 0.23 | Ref. | Ref. | Ref. |
| | 1 | 490 | 6 | 2147.91 | 0.28 | 1.21 (0.37–3.97) | 0.93 (0.28–3.05) | 0.71 (0.20–2.46) |
| | 2–5 | 722 | 45 | 3070.1 | 1.47 | 6.42 (2.55–16.18) | 3.95 (1.53–10.19) | 3.19 (1.22–8.35) |
| | >5 | 28 | 5 | 114.32 | 4.37 | 19.29 (5.58–66.63) | 10.95 (3.09–38.84) | 5.93 (1.49–23.69) |

BMI, body mass index; CI, confidence interval; HR, hazard ratio; IR, incidence rate; PY, person-years; Ref., reference category.
*Score was based on methylation of 10 CpGs (cg01612140, cg05575921, cg06126421, cg08362785, cg10321156, cg14975410, cg19572487, cg23665802, cg24704287 and cg25983901) using their respective first quartile values (cg08362785: using its highest quartile) among the ESTHER participants as the cutoff points to define aberrant methylation. Score 0–10 refer to simultaneously aberrant methylation at 0–10 CpGs.
†Incidence rate per 100 person-years.
‡Model 1: without adjustment.
§Model 2: adjusted for chronological age and sex.
‖Model 3: similar to model 2, additionally adjusted for smoking status, BMI, physical activity, alcohol consumption, systolic blood pressure, total cholesterol, hypertension and prevalent cardiovascular disease, diabetes and cancer at baseline.

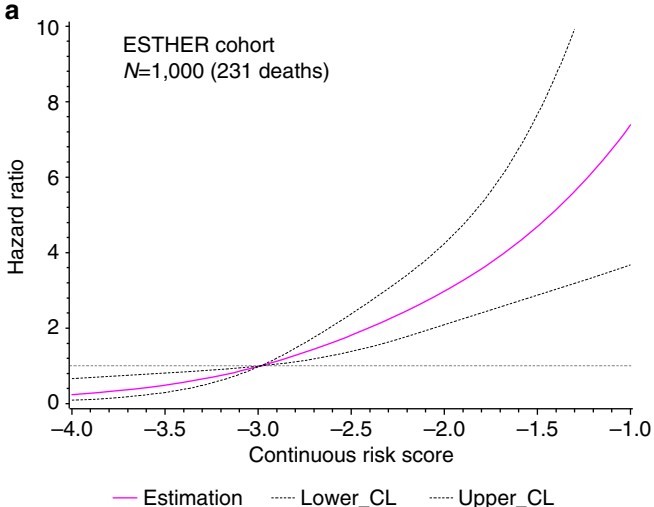

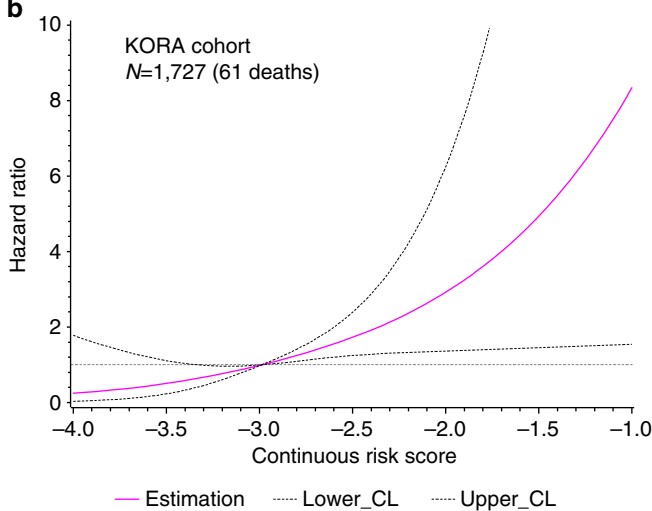

**Figure 2 | Dose–response relationships between continuous risk score and all-cause mortality.** (**a**) Dose–response curve in the ESTHER study ($N = 1,000$ (231 deaths)); (**b**) dose–response curve in the KORA study ($N = 1,727$ (61 deaths)).

survival curves were also obtained in the KORA cohort (Supplementary Fig. 5).

Table 5 presents the associations of score with all-cause and cause-specific mortality in the ESTHER cohort under consideration of the epigenetic age acceleration (determined by the algorithm of Hannum et al.[11]). The risk estimates of score for all three mortality outcomes were only very slightly attenuated by adjustment for the epigenetic age acceleration. On the contrary, HRs (95% CI) per 5 years of age acceleration dropped from 1.27 (1.10–1.46), 1.25 (0.98–1.59) and 1.34 (1.05–1.71), respectively, for all-cause, cancer and CVD mortality in the age- and sex-adjusted model to 1.08 (0.92–1.27), 1.15 (0.88–1.51) and 1.12 (0.85–1.48) in the full model. Similar results for the epigenetic age acceleration determined by the algorithm of Horvath et al.[12] are presented in Supplementary Table 6.

## Discussion

In this EWAS and subsequent validation based on approximately 1,900 older adults with up to 14 years of follow-up, we identified blood DNAm of 58 CpGs across 19 chromosomes to be associated with all-cause mortality. Although there is evidence that genes containing the identified CpGs are related to various types of common diseases, our study was the first to link DNAm of the vast majority of these genes to mortality in the general population. We additionally demonstrated that a risk score based on DNAm of ten identified CpGs was a very strong predictor for all-cause, CVD and cancer mortality, and we confirmed this finding in an independent cohort study. None of the newly identified CpGs overlapped with previously established ageing-related CpGs and the strong associations of score with mortality were also independent from the epigenetic clock.

Of the 58 identified CpGs, the top 1 locus showing the most significant association with mortality was cg05575921 in *AHRR*, followed by cg21161138 in *AHRR*, cg26963277 in *KCNQ1OT1*, cg19859270 in *GPR15*, cg03636183 in *F2RL3*, cg19572487 in *RARA* and cg06126421 in 6p21.33. All these CpGs (except cg26963277 in *KCNQ1OT1*) were also the top signals in previous EWASs on smoking[2]. In addition to the 22 CpGs identified to be associated with smoking in previous EWASs[2,24], another 26 of the 58 CpGs were also smoking-associated in the current study. Furthermore, even though a few other CpGs were found to be associated with alcohol consumption, diabetes or cancer, such as

**Table 4 | Associations of the risk score with cancer and CVD mortality in the ESTHER and KORA study.**

| Outcome | Study | Mortality score* | $N_{total}$ | Cases | PY | IR† | Model 1‡ | Model 2§ | Model 3‖ |
|---|---|---|---|---|---|---|---|---|---|
| | | | | | | | \multicolumn HR (95% CI) | | |
| Cancer mortality | ESTHER | 0 | 199 | 8 | 2690.69 | 0.30 | Ref. # | Ref. # | Ref. # |
| | | 1 | 242 | 17 | 3144.50 | 0.54 | | | |
| | | 2–5 | 426 | 31 | 5300.86 | 0.58 | 1.38 (0.82–2.34) | 1.24 (0.72–2.11) | 1.21 (0.68–2.15) |
| | | >5 | 131 | 22 | 1348.99 | 1.63 | 4.11 (2.31–7.30) | 3.12 (1.69–5.78) | 2.57 (1.27–5.21) |
| | KORA | 0 | 487 | 3 | 2163.01 | 0.14 | Ref. # | Ref. # | Ref. # |
| | | 1 | 490 | 1 | 2147.91 | 0.05 | | | |
| | | 2–5 | 722 | 16 | 3070.1 | 0.52 | 5.78 (1.93–17.31) | 4.28 (1.39–13.13) | 3.16 (1.01–9.85) |
| | | >5 | 28 | 2 | 114.32 | 1.75 | 19.42 (3.56–106.06) | 14.74 (2.6–83.69) | 5.74 (0.84–39.42) |
| CVD mortality | ESTHER | 0 | 199 | 4 | 2690.69 | 0.15 | Ref. # | Ref. # | Ref. # |
| | | 1 | 242 | 9 | 3144.50 | 0.29 | | | |
| | | 2–5 | 426 | 43 | 5300.86 | 0.81 | 3.69 (1.99–6.87) | 3.41 (1.82–6.40) | 4.00 (1.96–8.15) |
| | | >5 | 131 | 25 | 1348.99 | 1.85 | 9.04 (4.62–17.70) | 7.19 (3.54–14.62) | 9.12 (3.89–21.39) |
| | KORA | 0 | 487 | 2 | 2163.01 | 0.09 | Ref. # | Ref. # | Ref. # |
| | | 1 | 490 | 2 | 2147.91 | 0.09 | | | |
| | | 2–5 | 722 | 15 | 3070.1 | 0.49 | 5.23 (1.74–15.76) | 3.67 (1.19–11.35) | 4.89 (1.34–17.78) |
| | | >5 | 28 | 3 | 114.32 | 2.62 | 28.5 (6.38–127.36) | 19.18 (4.1–89.71) | 25.00 (3.99–156.43) |

CI, confidence interval; HR, hazard ratio; IR, incidence rate; PY, person-years; Ref., reference category.
*Score was based on methylation of 10 CpGs (cg01612140, cg05575921, cg06126421, cg08362785, cg10321156, cg14975410, cg19572487, cg23665802, cg24704287 and cg25983901) using their respective first quartile values (cg08362785: using its highest quartile) among the ESTHER participants as the cutoff points to define aberrant methylation. Score 0–10 refer to simultaneously aberrant methylation at 0–10 CpGs.
†Incidence rate per 100 person-years.
‡Model 1: without adjustment.
§Model 2: adjusted for chronological age and sex.
‖Model 3: similar to model 2, additionally adjusted for smoking status, body mass index, physical activity, alcohol consumption, systolic blood pressure, total cholesterol, hypertension and prevalent cardiovascular disease, diabetes and cancer at baseline. # Score=0–1 used as the reference group.

cg18181703 in *SOCS3* and cg26470501 in *BCL3*, most of them also showed associations with smoking exposure in our analyses (Supplementary Fig. 3). These findings suggest that tobacco smoking is the strongest factor leaving imprints on DNAm such that smoking rather than other common health risk factors accounts for the major burden of morbidity and mortality involving epigenetic programming. Regardless of the underlying mechanisms which remain to be elucidated in further research, it appears worthwhile pointing out that prevention of or intervention on smoking-related DNAm changes may provide major improvement in premature death prevention, given the reversibility of smoking-induced methylomic aberrations[29,30].

The current study highlighted several genes or genetic regions as attractive targets for further investigation. The chromosome region 19q13.3 harbours six mortality-related CpGs mapped to *BCL3*, *ERCC1*, *SLC1A5* and *PPP1R15A*. Although *ERCC1* methylation has been previously reported in lung and breast cancer[31,32], DNAm of *BCL3*, *SLC1A5* and *PPP1R15A* were first linked to health-related outcomes in our study. In light of the known gene functions of *BCL3* (pathogenesis of CVD and solid tumours)[33–35], *SLC1A5* (a glutamine transporter in various types of cancer development, progression and response to therapy)[36] and *PPP1R15A* (neurological and CVD pathophysiology, as well as obesity and insulin resistance in animal models)[37–39], it appears plausible that DNAm may play regulating roles in the development or progression of the respective diseases, which requires elucidation in future studies. This also applies to most of the other genes known to be related to specific diseases whose relationship to methylation-relevant outcomes were first disclosed in our study, such as DNAm of *SARS*, *VCAM1*, *KCNQ1OT1*, *MIR19A*, *SEMA7A*, *BCL3*, *PPP1R15A* and *PDE9A* for CVD, DNAm of *SQLE*, *MIR19A*, *MIR10A*, *SOCS3*, *CALR*, *BCL3* and *SLC1A5* for lung cancer, and DNAm of *ATL3*, *SHANK2* and *PPP1R15A* for neurological diseases. In addition, it is known that the chromosomal region 11p15.5 contains clusters of epigenetically regulated genes, for example, *KCNQ1* and *KCNQ1OT1*, which have been implicated in T2D[26,27]. We found

two mortality-related CpGs (cg07123182 and cg26963277) in *KCNQ1OT1* in 11p15.5. Of note, associations with prevalent T2D and mortality were also observed in the current study for cg23190089 in *SLC22A18AS* (on 11p15.4), a locus located ~198 kb downstream of cg26963277 in *KCNQ1OT1*. The chromosome region 11p15.5/4, along with *SLC1A5*, *SQLE* and *SOCS3* methylation that were suggested to be involved in T2D in both the current and previous studies[5,25,40–42], therefore appear to be attractive targets for diabetes investigation, and even for CVD given the biological functions of these genes and their methylation in CVD[40,43] and also the well-known causal relationship between diabetes and CVD. Similar to *SQLE*, *KCNQ1OT1* and *SOCS3*, which are involved in diabetes, CVD and various cancers[5,26,40,43–46], most identified genes are characterized by their relevance to multiple diseases, making them the most robust signals on an epigenome-wide scale, which may explain the extremely strong association of the risk score based on identified DNAm markers with all-cause mortality.

Compared with genetic variants related to longevity identified by GWAS, which typically show very small effect sizes of single SNPs, in particular in general population samples[47,48], the effect size of even single CpGs identified in the current EWAS were substantial, with HRs $\geq 1.17$ or $\leq 0.83$ per s.d. increase of methylation, resulting in the strong overall prediction when combining these CpGs in a risk score. To our knowledge, no comparably strong prediction of mortality based on genetic data has been identified, suggesting that epigenetic data might be more informative for mortality prediction than genetic data.

The recently established epigenetic clock (DNAm age) has received growing attention as an increasing number of studies have uncovered it to be a proxy of biological ageing[11–15] and thus potentially providing a measure for assessing health and mortality. Intriguingly, we targeted mortality-related DNAm changes and did not find any overlap with previously established CpGs that are used to determine the DNAm age[11,12]. Our findings are in line with evidence, suggesting that DNAm involved in ageing or health-related outcomes are mostly

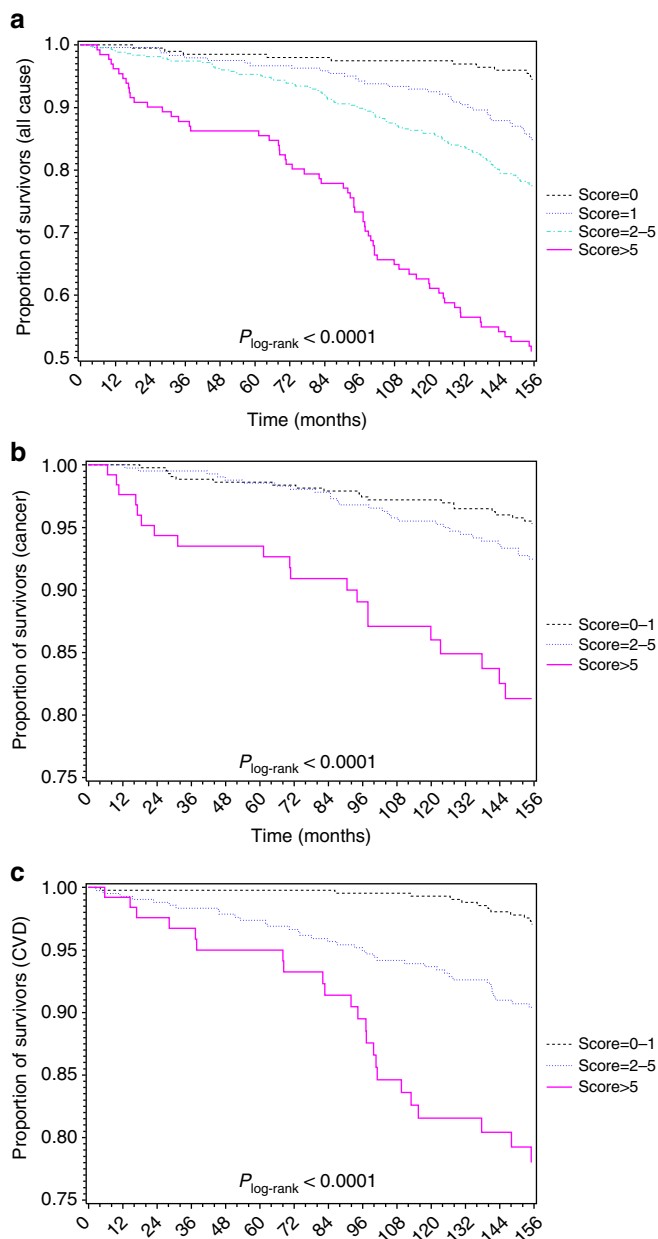

**Figure 3 | Kaplan–Meier estimates of survival by risk score in the ESTHER study (N = 1,000).** (**a**) Survival curves with respect to death from any causes; (**b**) survival curves with respect to death from cancer; (**c**) survival curves with respect to death from CVD. $P_{log-rank}$ was derived from log-rank test.

regulated by DNAm regions other than the established age-related DNAm[19–21]. The difference could also plausibly result from the fact that DNAm age was originally trained as precisely as possible to track chronological age and might thus be more indicative of natural ageing beyond the effect of disease, as exemplified by the much stronger association of DNAm age with mortality in oldest population (mean age 86.1 years)[15] to whom common chronic diseases, such as CVD and cancer, might not continue to pose predominant risks[49]. Given characteristics of the identified genes and of our study population (mean age 62 years) that is at high-risk age for suffering from major diseases, the currently identified DNAm regions might be more indicative of disease-related outcomes and mortality. Only one previous

study has also determined genome-wide methylomic mortality predictors, which were also distinct from the established ageing-associated sites[11,12] but also different from signatures discovered in our study. A plausible explanation is that this study was conducted in a very old population (mean age 90 + years), in which causes of death might be distinct from those observed in our study[49].

Lack of gene expression data hindered exploration of the roles of the identified DNAm sites in regulating the relevant gene expression. Diseases associated with the identified genes were determined based on a literature search. Whether and how DNAm of those genes are involved in development or progression of the described diseases needs to be elucidated by future multidisciplinary research. For example, genetic factors might potentially be involved in the observed methylation-related mortality and the interplay between genetic factors and these methylation markers warrants to be explored. In the analysis, we did not exclude probes that might be affected by known SNPs as annotated by 'Infinium HD Methylation SNP List' (http://support.illumina.com/array/array_kits/infinium_humanm ethylation450_beadchip_kit/downloads.html). We later retrieved data of 32 such SNPs for 24 identified CpGs in 581 ESTHER participants of the validation set. Only one SNP-CpG pair (rs524-cg03707168) showed a significant association. However, no association was observed between rs524 and all-cause mortality irrespective of controlling for DNAm of cg03707168, whereas the strong association of cg03707168 with mortality did not change when controlling for rs524. In addition, no interaction was detected between rs524 and cg03707168 in relation to mortality. Nevertheless, potential genetic variants, that is, methylation quantitative trait loci for the identified candidates, should be systematically assessed in further studies. Despite the overall large size of the study population, sample size limitations restricted the list of identified sites, which should be extended in future larger longitudinal studies. In addition, the effect sizes (that is, average methylation difference between survivors and deaths) of most identified loci are relatively small as illustrated in Fig. 1. Plausible reasons are that methylation levels were measured on average 8.2 years before dying and presumably stronger methylation difference restricted to specific causes of death are expected to be diluted in an analysis of all-cause mortality. Another limitation is that DNAm was quantified in whole blood samples. Even though we controlled for the effect of potential cell shift by adjustment for leukocyte composition estimated according to an established and commonly applied algorithm[50], residual confounding by leukocyte distribution cannot be ruled out. However, this would not diminish the value of the identified markers for mortality prediction, for which easy accessibility of blood samples is a major advantage. Finally, although we included a variety of covariates in the regression analyses, we cannot exclude the possibility that the observed associations between the identified methylation markers and mortality might be explained to some extent by incompletely controlled or uncontrolled confounding factors. For example, for smoking-related candidates, the observed associations might be partially confounded by imperfect controlling for smoking exposure or by potential confounders related to smoking. Despite its limitations, the prospective nature of the present study, the inclusion of large representative samples of participants from the general population, the long-term follow-up, the hypothesis-free approach with independent internal and external validation, as well as comprehensive adjustment for a variety of common risk factors in data analyses, are major strengths of the current study, which renders novel findings for future verification.

Our previous work using candidate gene approaches has demonstrated the potential use of mortality-related DNAm

**Table 5 | Associations of the risk score and epigenetic clock with all-cause and cause-specific mortality in the ESTHER study.**

| Outcome | Mortality score*/epigenetic clock† | HR (95% CI) | | |
|---|---|---|---|---|
| | | Model 1‡ | Model 2§ | Model 3‖ |
| All-cause mortality | 0 | Ref. | Ref. | Ref. |
| | 1 | 2.04 (1.11–3.75) | 2.02 (1.10–3.72) | 2.15 (1.09–4.21) |
| | 2–5 | 3.18 (1.81–5.59) | 3.07 (1.74–5.41) | 3.31 (1.75–6.28) |
| | >5 | 7.64 (4.21–13.85) | 7.18 (3.92–13.15) | 6.96 (3.46–14.01) |
| | Hannum Δage (per 5 years) | 1.27 (1.10–1.46) | 1.09 (0.94–1.27) | 1.08 (0.92–1.27) |
| Cancer mortality | 0–1 | Ref. | Ref. | Ref. |
| | 2–5 | 1.24 (0.72–2.11) | 1.19 (0.69–2.04) | 1.16 (0.65–2.06) |
| | >5 | 3.12 (1.69–5.78) | 2.89 (1.53–5.46) | 2.33 (1.12–4.84) |
| | Hannum Δage (per 5 years) | 1.25 (0.98–1.59) | 1.13 (0.87–1.46) | 1.15 (0.88–1.51) |
| CVD mortality | 0–1 | Ref. | Ref. | Ref. |
| | 2–5 | 3.41 (1.82–6.40) | 3.28 (1.74–6.18) | 3.85 (1.87–7.89) |
| | >5 | 7.19 (3.54–14.62) | 6.63 (3.19–13.78) | 8.47 (3.54–20.28) |
| | Hannum Δage (per 5 years) | 1.34 (1.05–1.71) | 1.12 (0.87–1.45) | 1.12 (0.85–1.48) |

CI, confidence interval; HR, hazard ratio; Ref., reference category.
*Score was based on methylation of 10 CpGs (cg01612140, cg05575921, cg06126421, cg08362785, cg10321156, cg14975410, cg19572487, cg23665802, cg24704287 and cg25983901) using their respective first quartile values (cg08362785: using its highest quartile) among the ESTHER participants as the cutoff points to define aberrant methylation. Score 0–10 refer to simultaneously aberrant methylation at 0–10 CpGs.
†The epigenetic clock estimated by the difference between DNA methylation age calculated according to Hannum's algorithm and chronological age.
‡Model 1: adjusted for age and sex.
§Model 2: similar to model 1, additionally adjusted for the epigenetic clock/risk score.
‖Model 3: similar to model 2, additionally adjusted for smoking status, body mass index, physical activity, alcohol consumption, systolic blood pressure, total cholesterol, hypertension and prevalent cardiovascular disease, diabetes and cancer at baseline.

markers, such as *F2RL3* and *AHRR*, for lung cancer and CVD risk prediction[8,9]. The clinical implications of other CpGs emerging from the present study for diagnosis, prognosis or even treatment of common diseases, in particular diabetes, CVD and cancer, warrant exploration by future studies. The methylation-based mortality risk score might be a useful tool for population stratification in disease screening and intervention, and its predictive value for ageing-related outcomes, such as frailty and dementia, is worthwhile investigating in future research.

## Methods

**Study population and data collection.** The EWAS and subsequent validation were conducted in the ESTHER study, an ongoing population-based cohort study conducted in Saarland, Germany. The ESTHER cohort, as previously described in detail[51], enroled 9,949 older adults (age 50–75 years) by their general practitioners during routine health check-ups between 2000 and 2002. The participants completed a standardized self-administered questionnaire and donated biological samples (blood, stool and urine) during baseline enrolment. Comprehensive medical data, such as the results of a physical assessment, medical diagnoses and drug prescriptions were additionally obtained from the general practitioner. Deaths during follow-up were identified through record linkage with population registries in Saarland. Information on the major cause of death was obtained from death certificates provided by the local health authorities and coded with ICD-10 codes. Deaths from CVD and malignant invasive cancers, respectively, were defined by ICD-10 codes I00-I99 and C00-C97 (excluding non-melanoma skin cancer (C44)).

Genome-wide DNAm measurements were performed in the baseline blood samples of two subsets of the ESTHER participants. Subset-I (discovery panel) consists of participants from a case–cohort study nested within 2,499 ESTHER participants who were consecutively recruited between October 2000 and March 2001, and had sufficient DNA available. Of the 2,499 participants, 406 participants who died during follow-up by March 2013 were the cases in the case–cohort design and 548 participants were randomly selected as the subcohort irrespective of death status during follow-up. The sampling fraction was thus 548/2,499 = 22%. Subset-II (validation panel) consists of 1,000 ESTHER participants who were recruited between July and October 2000, and who were non-overlapping with the case–cohort samples, among whom 231 deaths were ascertained during follow-up.

Replication in an independent cohort was performed in the KORA F4 study, a population-based cohort consisting of 3,080 participants (age 32–81 years) recruited between 2006 and 2008 from the region of Augsburg, Southern Germany[52,53]. The vital status of KORA participants was ascertained through population registries inside and outside the study area in December 2011. Causes of death were determined according to death certificates from the Regional Health Department and coded with ICD-9. A random baseline sample consisting of 1,727 participants were selected for methylation analysis, among whom 61 participants died.

All ESTHER and KORA F4 participants provided written informed consent. The ESTHER study was approved by the ethics committees of the University of Heidelberg and of the state medical board of Saarland, Germany. The KORA F4 study was approved by the Ethics Committee of the Bavarian Medical Association.

**Methylation assessment.** DNAm in whole blood was quantified using the Infinium HumanMethylation450K BeadChip (Illumina, Inc, San Diego, CA, USA) in both ESTHER and KORA F4. Details of methylation analysis in the ESTHER study have been reported previously[8,54]. According to the manufacturer's protocol, data were normalized to internal controls provided by Illumina (Illumina normalization). In data pre-processing, probes with detection *P*-value > 0.01, with missing values > 10%, probes targeting the sex chromosomes, cross-reactive probes and polymorphic CpGs[55] were excluded, leaving 430,363 CpGs for genome-wide screening. In the KORA study, data were pre-processed following the pipeline of Lehne *et al.*[56], probes with detection *P*-value (1 − *P*-value computed from the background model characterizing the probability that the target sequence signal was distinguishable from the negative controls) > 0.01 and missing values > 5% were removed, and quantile normalization was applied following stratification of the probe categories into six types, based on probe type and colour channel, using the R package limma[57]. Leukocyte composition was estimated using the algorithms of Houseman *et al.*[50] in both studies.

**Statistical analysis.** *Discovery and validation of mortality-related CpGs.* The ESTHER study populations were described separately in the discovery and validation panel with respect to major sociodemographic characteristics, lifestyle factors and prevalent diseases at baseline. An epigenome-wide screening for mortality-related CpGs was first carried out in the case–cohort samples, using weighted Cox regression models that account for the case–cohort sampling design by Barlow weighting (the inverse of the subcohort sampling fraction, 1/(548/2499))[58,59]. The models with methylation β-values as explanatory variables were adjusted for age, sex and batch effects. After correcting for multiple testing using the Benjamini–Hochberg approach, CpGs that reached genome-wide significance (FDR < 0.05) were entered into the validation phase, in which the associations with mortality were further analysed by multiple Cox regression adjusted for age, sex, batch effects, leukocyte composition[50], smoking status (never, former and current smoker), body mass index (kg m$^{-2}$), physical activity (inactive, low, medium/high), alcohol consumption (grams per day), systolic blood pressure (mmHg), total cholesterol level (mg dL$^{-1}$), and prevalence of hypertension, CVD, diabetes and cancer. CpGs with FDR < 0.05 in the validation panel were deemed as mortality-related loci. A flowchart of study design and data analysis is shown in Supplementary Fig. 1.

*Associations of risk factors with mortality-related CpGs.* To explore risk factors related to methylation associated with fatal endpoints, sociodemographic characteristics, lifestyle factors and prevalent diseases at baseline were assessed in relation to the methylation levels of the identified CpGs using mixed linear regression models in the validation panel, with batch as random effect, methylation β-value as the dependent variable and independent variables including age, sex, smoking status (never, former and current smoker), body mass index

(underweight/normal weight, overweight and obesity), physical activity (inactive, low and medium/high), alcohol consumption (grams per day) and prevalent hypertension, diabetes, CVD and cancer, again controlling for leukocyte composition[50]. Multiple testing was again corrected for by the Benjamini–Hochberg approach (FDR < 0.05).

*Mortality risk score.* To develop a DNAm-based mortality risk score, we applied the LASSO Cox regression[60] with regularization parameter chosen by tenfold cross-validation following the 'one standard error' rule[61,62], selecting candidates among the identified CpGs. The associations of the score with all-cause, CVD and cancer mortality were assessed first in the validation subset of the ESTHER cohort and then in the independent KORA cohort using multiple Cox regression models, adjusted for the covariates listed above (Supplementary Fig. 1). All analyses were then repeated in men and women separately. In addition, to compare the predictive value of score with that of recently established methylomic predictors of 'epigenetic age acceleration' (that is, $\Delta$age = DNAm age – chronological age), we assessed the associations of both score and $\Delta$age with all-cause mortality simultaneously. DNAm age was calculated according to two commonly applied algorithms introduced by Hannum *et al.*[11] and Horvath *et al.*[12].

The proportional hazards assumption was assessed by martingale-based residuals[63]. No violations were detected. The LASSO regression analyses were conducted using the R-package 'glmnet'[61]. All other statistical analyses in the ESTHER study were carried out in SAS 9.4 (SAS Institute, Cary, NC) and the analyses in the KORA study were conducted in R (version 3.2.3).

**Code availability.** SAS codes for statistical analysis are available upon request.

**Data availability.** The data that support the findings of this study are available on reasonable request from the corresponding author (Y.Z.). The data are not publicly available due to restrictions of informed consent.

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

## Acknowledgements

The ESTHER study was supported by the Baden-Württemberg State Ministry of Science, Research and Arts (Stuttgart, Germany), the Federal Ministry of Education and Research (Berlin, Germany) and the Federal Ministry of Family Affairs, Senior Citizens, Women and Youth (Berlin, Germany). The KORA study was initiated and financed by the Helmholtz Zentrum München–German Research Center for Environmental Health, which is funded by the German Federal Ministry of Education and Research (BMBF) and by the State of Bavaria. The research leading to these results has received funding from the European Union Seventh Framework Programme (FP7/2007-2013) under grant agreement number 603288 (SysVasc). The sponsors had no role in the study design, in the collection, analysis and interpretation of data and preparation, review or approval of the manuscript.

## Author contributions

Y.Z. conceived the study, carried out the data analyses, interpreted the data and drafted the manuscript. R.W. conducted data analyses in the KORA study. J.H. computed leukocyte composition in the ESTHER study. L.P.B. contributed to the design of the study. K.S. was responsible for coordination of the ESTHER study. B.S. and B.H. were responsible for coordination of follow-up and work-up of follow-up data of the ESTHER study. M.W. and A.P. are the investigators of the KORA study. H.B. conducted the ESTHER study and contributed to all aspects of this work. All authors contributed to revision of the manuscript and approved the final version for submission.
