## [Peer Review File · Nature Communications]

Reviewers' Comments:

Reviewer #1 (Remarks to the Author)

This is a well-written, concise and concise manuscript in which you develop a risk-stratification compound epigenetic biomarker (using 10 CpGs) for all-cause mortality. These CpGs came from an EWAS with $n \geq 1000$ in original and replication cohorts together with a validation stage before analysis of the second cohort. Genes associated with risk for mortality had been associated with specific environments (mainly smoking) and with a number of chronic diseases.

While there have been a very small number of other potential biomarkers of mortality, your study is not based on methylation age and is therefore original and will be of wider interest to the scientific and medical community. Your data was based on the HM450 array and a sound statistical analysis. Conclusions were valid and based on validated and replicated data.

I have 3 minor questions for you to address:

1. Please discuss the effect size ($< 5\%$) for significant CpG and what the implications of this are
2. As you didn't remove probes influenced by known SNPs (which I am not against), do you think that some of the risk is genetic rather than epigenetic in nature. It would be good to test this hypothesis if you have genetic data.
3. Do you have the data on medicaor family history with which to augment your risk model?

Reviewer #2 (Remarks to the Author)

This is an interesting study that, to my knowledge, provides the first associations between individuals methylation CpGs and mortality.

Major comments:

1. I found the study design slightly confusing. Why did the authors' not include covariates in the discovery panel but did in the replication analyses? A flow chart figure of the study design would be helpful.
2. There is quite a wide range of ages in the study, including many younger adults. Given that deaths in the younger age range will likely be due to accidents or specific cancer sub-types, I wonder what information the censored observations from this group will be adding. Did the authors' consider running a models in the older individuals only?
3. The discovery EWAS did not control for white cell counts. Why not?
4. The risk score analysis is nice although the categories are not that intuitive. Did the authors' see similar trends in a continuous weighted score?
5. Even after adjusting for smoking status, the majority of the top hits have been identified in previous EWAS studies of smoking. Does this imply imperfect control for smoking status and association via confounding? This needs to be discussed.

Response to Referees

We appreciate the insightful and constructive comments from the two reviewers and have revised the manuscript to address each of the comments. In the following paragraphs, the reviewers' comments are highlighted in bold text. In addition, we modified the abstract to meet the journal requirements.

Reviewer #1 (Remarks to the Author):

This is a well-written, concise and concise manuscript in which you develop a risk-stratification compound epigenetic biomarker (using 10 CpGs) for all-cause mortality. These CpGs came from an EWAS with $n \geq 1000$ in original and replication cohorts together with a validation stage before analysis of the second cohort. Genes associated with risk for mortality had been associated with specific environments (mainly smoking) and with a number of chronic diseases.

While there have been a very small number of other potential biomarkers of mortality, your study is not based on methylation age and is therefore original and will be of wider interest to the scientific and medical community. Your data was based on the HM450 array and a sound statistical analysis. Conclusions were valid and based on validated and replicated data.

I have 3 minor questions for you to address:

1. Please discuss the effect size (<5%) for significant CpG and what the implications of this are

Response: We added a paragraph on the effect sizes for significant CpGs in the discussion as suggested (page 13, para 2, as follows)

"Compared to genetic variants related to longevity identified by GWAS, which typically show very small effect sizes of single SNPs, in particular in general population samples^{47,48}, the effect size of even single CpGs identified in the current EWAS were substantial, with HRs ≥ 1.17 or ≤ 0.83 per SD increase of methylation, resulting in the strong overall prediction when combining these CpGs in a risk score. To our knowledge, no comparably strong prediction of mortality based on genetic data has been identified, suggesting that epigenetic data might be more informative for mortality prediction than genetic data."

2. As you didn't remove probes influenced by known SNPs (which I am not against), do you think that some of the risk is genetic rather than epigenetic in nature. It would be good to test this hypothesis if you have genetic data.

Response: We fully agree with the reviewer. However, most candidates are newly identified. Findings regarding SNPs related to the identified CpGs are very sparse, but very worthwhile disclosing. Unfortunately, genetic data are not available for ESTHER participants included in the current study. We now add this important point in the discussion (page 14, para 2, as follows).

"Whether and how DNAm of those genes are involved in development or progression of the described diseases needs to be elucidated by future multidisciplinary research. For example, genetic factors might potentially be involved in the observed methylation-related mortality, and the interplay between genetic factors and these methylation markers warrants to be explored."

3. Do you have the data on medicaor family history with which to augment your risk model?

Response: In ESTHER study, we collected information on family history of common chronic diseases (occurred in participants' parents, siblings, and children), including family history of diabetes, cancer, and cardiovascular disease (CVD, including myocardial infarction and stroke). We calculated the associations of these three types of family history of diseases with mortality from CVD, cancer and any causes. The results are listed in the following Table 1. None of the family history variables showed a significant association with fatal outcomes. Additional adjustment for them in the model estimating the association between risk score and mortality also led to very limited changes of the risk estimates as listed in the following Table 2. We therefore decided not to include this information in the manuscript.

Table 1. Associations between family history of chronic diseases and mortality in validation panel (ESTHER study, n=1000)

Family history of diseases	Hazard ratio (95% CI)		
	All-cause mortality	CVD mortality	Cancer mortality
Diabetes (n=381)	0.78 (0.59 – 1.03)	1.11 (0.71 – 1.75)	
CVD (n=181)	1.11 (0.80 – 1.54)	1.33 (0.79 – 2.25)	
Cancer (n=446)	1.06 (0.82 – 1.38)		1.33 (0.85 -2.08)

Table 2. Associations of risk score with mortality with and without adjustment for family history of disease (ESTHER study, n=1000)

Outcome	Mortality score ^a	Hazard ratio (95% CI)	
		Model 1 ^b	Model 2 ^c
All-cause mortality	0	Ref.	Ref.
	1	2.55 (1.39 – 4.68)	2.65 (1.42 – 4.96)
	2-5	3.93 (2.25 – 6.86)	4.15 (2.33 – 7.40)
	>5	10.89 (6.13 – 19.35)	10.42 (5.71 – 19.00)
CVD mortality	0	Ref.	Ref.
	1		
	2-5	3.69 (1.99 – 6.87)	3.62 (1.94 – 6.75)
	>5	9.04 (4.62 – 17.70)	8.59 (4.35 – 16.98)
Cancer mortality	0	Ref.	Ref.
	1		
	2-5	1.38 (0.82 – 2.34)	1.41 (0.84 – 2.40)
	>5	4.11 (2.31 – 7.30)	4.07 (2.28 – 7.29)

^aScore was based on methylation of 10 CpGs (cg01612140, cg05575921, cg06126421, cg08362785, cg10321156, cg14975410, cg19572487, cg23665802, cg24704287, cg25983901); Score 0-10 refer to simultaneously aberrant methylation at 0 to 10 CpGs; ^bModel 1: without adjustment; ^cModel 2: adjusted for family history of diabetes, cancer, and CVD in analysis for all-cause mortality; adjusted for family history of CVD in analysis for CVD mortality; adjusted for family history of cancer in analysis for cancer mortality.

Reviewer #2 (Remarks to the Author):

This is an interesting study that, to my knowledge, provides the first associations between individuals methylation CpGs and mortality.

Major comments:

1. I found the study design slightly confusing. Why did the authors' not include covariates in the discovery panel but did in the replication analyses? A flow chart figure of the study design would be helpful.

Response: We regret the confusion. As done in other epigenetic studies in this field¹⁻³, which commonly adjusted for basic confounding factors, including age, sex, and batch effects, we included those factors as covariates in the genome-wide screening phase. Limited adjustment in the EWAS screening phase also allows more potential biologically relevant candidates to enter the validation phase in which the most thorough correction for potential confounding factors and multiple testing were employed. A flowchart for study design and data analysis has now been added as suggested (Supplementary Figure 1).

2. There is quite a wide range of ages in the study, including many younger adults. Given that deaths in the younger age range will likely be due to accidents or specific cancer sub-types, I wonder what information the censored observations from this group will be adding. Did the authors' consider running a models in the older individuals only?

Response: As described in the Methods section (page 17, para 1, line 3), the ESTHER study enrolled participants 50-75 years of age. Table 1 shows that approximately 70% of the ESTHER participants were ≥ 60 years. We now change the presentation of the lowest age category in Table 1 to make this explicit. Only the KORA participants (used for validation of the risk score) had a broad age range (31-82 years), but the average age of both cohorts are similar: KORA vs. ESTHER, 61 vs. 62 years (page 6, para 1, line 13). Therefore, the model was developed in older adults (ESTHER cohort) and was also validated basically in older adults (KORA cohort). When repeating the analyses in the validation samples in only older adults (≥ 60 years) as suggested by the reviewer, results essentially remained the same. We now explicitly add this important information (page 9, para 1, as follows).

"Analyses restricted to only older participants (≥ 60 years) yielded essentially the same risk estimates, e.g. HRs (95% CI) were 2.14 (1.02-4.47), 3.38 (1.68-6.80), and 7.44 (3.50-15.84), respectively, for scores of 1, 2-5, and 5+ vs. score=0."

3. The discovery EWAS did not control for white cell counts. Why not?

Response: As addressed in Response to the 1st comment from Reviewer #2, we aimed to let more potential biologically relevant candidates enter the validation phase, in which we performed thorough correction for potential confounding factors, including WBC counts.

4. The risk score analysis is nice although the categories are not that intuitive. Did the authors' see similar trends in a continuous weighted score?

Response: We computed a continuous risk score which is a linear combination of methylation values of each 10 CpG multiplied by its corresponding LASSO regression coefficients, and then modelled the dose-response relationships between this continuous risk score and all-cause mortality. A monotonous increase in mortality with increasing continuous risk score was observed in both ESTHER and KORA cohorts. We included this information in the text (page 9, para 1 as follows) and Figure 2.

“In addition, a continuous risk score was computed through linear combination of LASSO regression coefficient weighted methylation values of the 10 CpGs (the combination formula is presented in Supplementary Fig. 1). A similar trend that mortality monotonously increased with increasing continuous risk score was observed in both the ESTHER [risk score ranged from -3.92 to -0.72; median (IQR), -2.70 (-2.98 - -2.35)] and the KORA cohorts [risk score ranged from -4.40 to -1.51; median (IQR), -3.15 (-3.41 - -2.86)]. Figure 2 shows the corresponding dose-response relationships derived from restricted cubic spline regression with adjustment for all the covariates again.”

5. Even after adjusting for smoking status, the majority of the top hits have been identified in previous EWAS studies of smoking. Does this imply imperfect control for smoking status and association via confounding? This needs to be discussed.

Response: We thank the reviewer for this suggestion. Smoking status (coded either as smokers/non-smokers, or as current smokers/former smokers/never smokers as in the current study) is the most commonly controlled form for smoking exposure in the association analysis of mortality. But we fully agree with the reviewer and added a discussion as suggested (page 15, para 1, as follows).

“Finally, although we included a variety of covariates in the regression analyses, we cannot exclude the possibility that the observed associations between the identified methylation markers and mortality might be explained to some extent by incompletely controlled or uncontrolled confounding factors. For example, for smoking-related candidates, the observed associations might be partially confounded by imperfect controlling for smoking exposure or by potential confounders related to smoking.”

References

- 1 Fasanelli, F. *et al.* Hypomethylation of smoking-related genes is associated with future lung cancer in four prospective cohorts. *Nat Commun.* **6**, 10192 (2015).
- 2 Teschendorff, A. E. *et al.* Correlation of Smoking-Associated DNA Methylation Changes in Buccal Cells With DNA Methylation Changes in Epithelial Cancer. *JAMA Oncol.* **1**, 476-485 (2015).
- 3 Shenker, N. S. *et al.* Epigenome-wide association study in the European Prospective Investigation into Cancer and Nutrition (EPIC-Turin) identifies novel genetic loci associated with smoking. *Hum Mol Genet.* **22**, 843-851 (2013).

Reviewers' Comments:

Reviewer #1 (Remarks to the Author)

Thank you for your response. I am happy with all replies apart from:

Comment 1. I was referring to the 5% methylation effect size and its comparison to other EWAS data. Can you please comment on this too?

Comment 2. In this comment I was referring to CpG probes that could be affected by SNPs, which are annotated in the Infinium annotation. Please comment on this.

Response to Referees

We appreciate the reviewer's further comments and have revised the manuscript to address the corresponding questions. In the following paragraphs, the reviewer's comments are highlighted in bold text.

Reviewer #1 (Remarks to the Author):

Thank you for your response. I am happy with all replies apart from:

Comment 1. I was referring to the 5% methylation effect size and its comparison to other EWAS data. Can you please comment on this too? (Previous comments: Please discuss the effect size (<5%) for significant CpG and what the implications of this are?)

Response: If effect size refers to phenotype related methylation alteration, i.e. average difference in DNAm level of each CpG site between survivors and deaths in the current study, 57 of the 58 CpGs showed effect sizes <5% (except cg05575921) as listed in the following Table i. Such magnitude of effect size is not uncommon in EWAS, particularly in longitudinal studies with methylation levels measured in samples collected several years before events occurred (on average more than 8 years before dying in the ESTHER participants). For example, in an EWAS conducted by Guarrera et al. addressing blood DNA methylation profiles related to myocardial infarction risk, the effect size for the validated top one candidate was 0.016 based on a meta-analysis of two nested case-control samples¹. In the EWAS investigating type 2 diabetes associated methylation, effect sizes for pyrosequencing-validated top five loci were 0.05, -0.02, 0, -0.02, and 0.02². It could be assumable that the effect sizes of the candidate loci would be more distinct when approaching the occurrence of events, which is consistent with observations that effect sizes reported by cross-sectional studies tend to be larger^{3,4}. Furthermore, as the outcome in our study is all-cause mortality, stronger methylation differences associated with specific causes of deaths, such as those observed for smoking-related sites in relation to lung cancer showed in the same study population in our previous investigation⁵, might be diluted by the absence of such differences in relation to other causes of death. We now add these important points in the discussion (page 14, para 1) as follows:

“In addition, the effect sizes (i.e. average methylation difference between survivors and deaths) of most identified loci are relatively small as illustrated in Figure 1. Plausible reasons are that methylation levels were measured on average 8.2 years before dying and presumably stronger methylation difference restricted to specific causes of death are expected to be diluted in an analysis of all-cause mortality.”

Table i. Methylation levels of 58 CpGs and risk estimates for all-cause mortality

CpGs	Mean methylation		Effect size	SD	Cox regression		
	survivors	deaths			Regression coefficients	HR	95% CI
cg08362785	0.6505	0.6719	-0.0214	0.0351	13.0519	0.63	(0.51 - 0.78)
cg15459165	0.4001	0.4190	-0.0189	0.0427	11.8763	0.60	(0.47 - 0.77)
cg25491402	0.6264	0.6442	-0.0178	0.0437	9.7574	0.65	(0.47 - 0.90)
cg04987734	0.4217	0.4387	-0.0171	0.0427	4.9723	0.81	(0.70 - 0.94)
cg23842572	0.7403	0.7548	-0.0145	0.0393	7.2647	0.75	(0.62 - 0.91)
cg16503724	0.4582	0.4708	-0.0126	0.0386	6.6198	0.77	(0.64 - 0.94)
cg25193885	0.8523	0.8597	-0.0073	0.0206	12.3266	0.78	(0.65 - 0.93)
cg19459791	0.7375	0.7444	-0.0069	0.0383	4.9251	0.83	(0.71 - 0.97)
cg01140244	0.3014	0.3006	0.0008	0.0316	11.5692	0.69	(0.54 - 0.89)
cg25285720	0.9557	0.9519	0.0038	0.0177	-12.4045	1.25	(1.06 - 1.46)
cg02657160	0.8350	0.8298	0.0052	0.0253	-7.7029	1.22	(1.07 - 1.38)
cg14855367	0.8234	0.8171	0.0063	0.0366	-5.6444	1.23	(1.08 - 1.40)
cg19859270	0.8801	0.8735	0.0066	0.0206	-13.406	1.32	(1.13 - 1.54)
cg19266329	0.5943	0.5874	0.0070	0.0368	-7.77	1.33	(1.14 - 1.55)
cg12513616	0.4564	0.4471	0.0093	0.0382	-4.1363	1.17	(1.01 - 1.36)
cg07123182	0.9415	0.9319	0.0096	0.0230	-10.142	1.26	(1.11 - 1.44)
cg07626482	0.3066	0.2965	0.0100	0.0296	-5.9218	1.19	(1.03 - 1.38)
cg23079012	0.9644	0.9525	0.0119	0.0311	-4.7219	1.16	(1.04 - 1.28)
cg18181703	0.4631	0.4511	0.0120	0.0397	-5.4717	1.24	(1.07 - 1.44)
cg14817490	0.3133	0.3011	0.0122	0.0528	-3.3257	1.19	(1.00 - 1.42)
cg15342087	0.8221	0.8097	0.0124	0.0319	-4.8923	1.17	(1.01 - 1.36)
cg06905155	0.7129	0.6994	0.0135	0.0363	-4.9148	1.20	(1.05 - 1.36)
cg00310412	0.4980	0.4840	0.0140	0.0362	-6.2809	1.26	(1.07 - 1.47)
cg14975410	0.4796	0.4656	0.0140	0.0507	-3.5727	1.20	(1.04 - 1.38)
cg19572487	0.5225	0.5067	0.0158	0.0564	-4.1621	1.26	(1.07 - 1.49)
cg14085840	0.3366	0.3206	0.0160	0.0404	-9.2316	1.45	(1.10 - 1.91)
cg24397007	0.6170	0.6008	0.0161	0.0439	-5.6592	1.28	(1.08 - 1.53)
cg26286961	0.7913	0.7752	0.0161	0.0512	-4.7263	1.27	(1.10 - 1.47)
cg00285394	0.5868	0.5701	0.0167	0.0543	-3.3118	1.20	(1.05 - 1.36)
cg26963277	0.8906	0.8735	0.0171	0.0376	-7.1474	1.31	(1.14 - 1.49)
cg20732076	0.1648	0.1472	0.0175	0.0364	-6.2152	1.25	(1.05 - 1.50)
cg26470501	0.5200	0.5024	0.0176	0.0385	-4.7556	1.20	(1.04 - 1.39)
cg08546016	0.1676	0.1497	0.0180	0.0390	-8.3478	1.39	(1.13 - 1.70)
cg23190089	0.1306	0.1125	0.0182	0.0324	-10.4437	1.40	(1.08 - 1.82)
cg01572694	0.5132	0.4947	0.0185	0.0528	-5.8867	1.36	(1.12 - 1.67)
cg21161138	0.7008	0.6823	0.0185	0.0479	-4.3571	1.23	(1.05 - 1.44)
cg03725309	0.1869	0.1671	0.0199	0.0364	-7.9824	1.34	(1.10 - 1.62)
cg01406381	0.1238	0.1029	0.0209	0.0367	-11.465	1.52	(1.19 - 1.95)
cg25607249	0.1477	0.1268	0.0209	0.0390	-8.8837	1.41	(1.10 - 1.81)

Table i. continued

CpGs	Mean methylation		Effect size	SD	Cox regression		
	survivors	deaths			Regression coefficients	HR	95% CI
cg27241845	0.6411	0.6200	0.0211	0.0540	-3.8602	1.23	(1.06 - 1.44)
cg11341610	0.1977	0.1762	0.0215	0.0419	-6.0804	1.29	(1.04 - 1.59)
cg25763716	0.2522	0.2285	0.0237	0.0433	-5.9104	1.29	(1.02 - 1.63)
cg25983901	0.5209	0.4967	0.0242	0.0473	-3.7191	1.19	(1.02 - 1.40)
cg07986378	0.6301	0.6055	0.0246	0.0586	-4.0617	1.27	(1.03 - 1.57)
cg05492306	0.2664	0.2390	0.0274	0.0464	-7.1104	1.39	(1.07 - 1.80)
cg10321156	0.4433	0.4158	0.0274	0.0743	-2.4444	1.20	(1.02 - 1.42)
cg23665802	0.3372	0.3097	0.0275	0.0481	-6.8111	1.39	(1.13 - 1.71)
cg18550212	0.2918	0.2643	0.0276	0.0552	-8.1521	1.57	(1.22 - 2.01)
cg01612140	0.3803	0.3526	0.0277	0.0522	-6.4106	1.40	(1.14 - 1.72)
cg25189904	0.4571	0.4274	0.0296	0.0821	-2.0488	1.18	(1.02 - 1.38)
cg13854219	0.3337	0.3040	0.0297	0.0626	-6.5713	1.51	(1.05 - 2.17)
cg12510708	0.2884	0.2582	0.0302	0.0565	-5.0597	1.33	(1.06 - 1.67)
cg03636183	0.6731	0.6422	0.0309	0.0720	-3.2718	1.27	(1.06 - 1.51)
cg03707168	0.2934	0.2609	0.0325	0.0607	-4.1544	1.29	(1.02 - 1.63)
cg24704287	0.3525	0.3167	0.0358	0.0613	-4.3657	1.31	(1.06 - 1.61)
cg26709988	0.4672	0.4309	0.0363	0.0875	-7.6461	1.95	(1.29 - 2.94)
cg06126421	0.6615	0.6165	0.0449	0.0812	-3.5103	1.33	(1.10 - 1.60)
cg05575921	0.8244	0.7708	0.0536	0.1126	-3.6864	1.51	(1.25 - 1.84)

Comment 2. In this comment I was referring to CpG probes that could be affected by SNPs, which are annotated in the Infinium annotation. Please comment on this.

Response: We thank the reviewer's comments and clarification. We searched 'Infinium HD Methylation SNP List' (available at: http://support.illumina.com/array/array_kits/infinium_humanmethylation450_beadchip_kit/downloads.html) for SNPs that could potentially affect the identified 58 probes. As shown in the following Table ii, 27 of 58 probes have total of 45 such SNPs. The vast majority of the 45 SNPs have very low minor allele frequency (MAF; range, 0.000220 - 0.093724; the 5th column of Table ii) according to Illumina's list. Meanwhile, we were able to obtain OncoArray data for 581 ESTHER participants in the validation set (deaths=112). We extracted genotype data for the 32 SNPs that are covered by OncoArray chip (the 6th column of Table ii), checked their MAF among the current study population (the 7th column of Table ii), and analyzed the associations for the CpG-SNP pairs by Wilcoxon test. Only cg03707168-rs524 showed a significant association. We then estimated individual and joint associations of cg03707168 and rs524 with all-cause mortality, adjusting for covariates as described in the main text. The results are listed in the following Table iii: cg03707168 was strongly associated with all-cause mortality irrespective of adjustment for rs524, whereas rs524 was not associated with all-cause mortality. And no interaction was detected between cg03707168 and rs524 ($p=0.053$). We add this information in the discussion (page 13, para 2; page 14, para 1) as follows:

“In the analysis we did not exclude probes that might be affected by known SNPs as annotated by ‘Infinium HD Methylation SNP List’ (http://support.illumina.com/array/array_kits/infinium_humanmethylation450_beadchip_kit/download.s.html). We later retrieved data of 32 such SNPs for 24 identified CpGs in 581 ESTHER participants of the validation set. Only one SNP-CpG pair (rs524-cg03707168) showed a significant association. However, no association was observed between rs524 and all-cause mortality irrespective of controlling for DNAm of cg03707168, whereas the strong association of cg03707168 with mortality did not change when controlling for rs524. And no interaction was detected between rs524 and cg03707168 in relation to mortality. Nevertheless, potential genetic variants, i.e. methylation quantitative trait loci (meQTLs) for the identified candidates, should be systematically assessed in further studies.”

Table ii. A list of SNPs that could potentially impact methylation of the identified probes

CpG site	SNP	Distance (bp)	CHR	MAF (illumina)	Available on oncoArray	MAF (ESTHER)	P-value ^a (CpG-SNP pair)
cg03725309	rs146139983	19	1	0.0014	√	0	
cg13854219	rs151283647	3	1	0.0018	√	0	
cg13854219	rs140507371	19	1	0.0005	√	0	
cg15459165	rs146589452	40	1	0.0005			
cg24397007	rs188572475	8	2	0.0005	√	0	
cg14855367	rs191469675	13	3	0.0009	√	0	
cg14855367	rs75814705	32	3	0.5			
cg14975410	rs113326153	34	3	0.013266	√	0.0044	0.49
cg14975410	rs181160598	40	3	0.0005	√	0.0009	0.91
cg19859270	rs202120393	1	3	0.001			
cg19859270	rs145764013	7	3	0.001			
cg12513616	rs13436787	5	5	0.0664	√	0.0089	0.74
cg12513616	rs6878985	49	5	0.185506	√	0.0191	0.87
cg14817490	rs150632254	0	5	0.0005	√	0	
cg14817490	rs139711357	19	5	0.0009	√	0	
cg21161138	rs111875483	1	5	0.5			
cg21161138	rs189907270	2	5	0.0005	√	0	
cg01612140	rs142626437	2	6	0.0005	√	0	
cg12510708	rs190265432	37	7	0.0005	√	0	
cg00285394	rs73704507	2	8	0.022495	√	0	
cg26286961	rs79865370	48	8	0.093724	√	0.0957	0.73
cg25193885	rs188147583	1	11	0.0014			
cg25193885	rs150761248	48	11	0.0009	√	0	
cg00310412	rs185866394	1	15	0.0009	√	0	
cg26709988	rs12925986	24	16	0.257091			
cg26709988	rs182746533	40	16	0.0014			
cg01572694	rs150632354	34	17	0.0018	√	0.0018	0.59
cg18181703	rs145006362	1	17	0.000228			
cg18181703	rs149892338	20	17	0.001574			
cg03636183	rs200274536	34	19	0.0005	√	0	

Table ii. continued

CpG site	SNP	Distance (bp)	CHR	MAF (illumina)	Available on oncoArray	MAF (ESTHER)	P-value ^a (CpG-SNP pair)
cg03707168	rs202191597	18	19	0.0005	√	0	
cg03707168	rs524	40	19	0.352951	√	0.2883	<0.0001
cg03707168	rs35007147	50	19	0.000555			
cg05492306	rs3212930	16	19	0.15765	√	0.2317	0.73
cg05492306	rs3212929	34	19	0.128517	√	0.0017	0.45
cg05492306	rs146183838	38	19	0.0069	√	0	
cg07626482	rs183348736	31	19	0.0005	√	0	
cg11341610	rs143361476	2	19	0.00022			
cg11341610	rs192388573	13	19	0.0005	√	0	
cg26470501	rs193046509	31	19	0.0014	√	0	
cg26470501	rs185267410	32	19	0.0005	√	0	
cg25491402	rs190354037	1	21	0.0005	√	0	
cg25491402	rs145182689	51	21	0.0073	√	0.0044	
cg08362785	rs150793465	3	22	0.0003			
cg08362785	rs199683643	34	22	0.0005	√	0	

^aP-value derived from Wilcoxon rank test.

Table iii. Associations of cg03707168 and rs524 with all-cause mortality

Marker	genotype	HR (95% CI) ^a		
		Model 1	Model 2	Model 3
cg03707168 ^b	-	1.53 (1.21 – 1.95)		1.56 (1.23 – 1.99)
rs524	CC		Ref.	Ref.
	CT		0.84 (0.56 – 1.27)	0.79 (0.52 – 1.20)
	TT		0.95 (0.46 – 1.94)	0.75 (0.36 – 1.54)

^aadjusted for age, sex, smoking status, BMI, physical activity, alcohol consumption, systolic blood pressure, total cholesterol, hypertension, and prevalent cardiovascular disease, diabetes, and cancer at baseline, and batch effect for methylation measurement. ^bHR (95% CI) was calculated for per standard deviation decrease in methylation of cg03707168.

- Guarrera, S. *et al.* Gene-specific DNA methylation profiles and LINE-1 hypomethylation are associated with myocardial infarction risk. *Clin Epigenetics*. **7**, 133 (2015).
- Kulkarni, H. *et al.* Novel epigenetic determinants of type 2 diabetes in Mexican-American families. *Hum Mol Genet*. **24**, 5330-5344 (2015).
- Dayeh, T. *et al.* Genome-wide DNA methylation analysis of human pancreatic islets from type 2 diabetic and non-diabetic donors identifies candidate genes that influence insulin secretion. *PLoS Genet*. **10**, e1004160 (2014).
- Soriano-Tarraga, C. *et al.* Epigenome-wide association study identifies TXNIP gene associated with type 2 diabetes mellitus and sustained hyperglycemia. *Hum Mol Genet*. **25**, 609-619 (2016).
- Zhang, Y. *et al.* Comparison and combination of blood DNA methylation at smoking-associated genes and at lung cancer-related genes in prediction of lung cancer mortality. *Int J Cancer*. **139**, 2482-2492 (2016).

Reviewers' Comments:

Reviewer #1 (Remarks to the Author)

I am satisfied that you have addressed my two remaining queries